# A Novel Multiplex qRT-PCR Assay to Detect SARS-CoV-2 Infection: High Sensitivity and Increased Testing Capacity

**DOI:** 10.3390/microorganisms8071064

**Published:** 2020-07-17

**Authors:** Sara Petrillo, Giovanna Carrà, Paolo Bottino, Elisa Zanotto, Maria Chiara De Santis, Jean Piero Margaria, Alessandro Giorgio, Giorgia Mandili, Miriam Martini, Rossana Cavallo, Davide Barberio, Fiorella Altruda

**Affiliations:** 1Department of Molecular Biotechnology and Health Sciences, University of Torino, 10126 Torino, Italy; mariachiara.desantis@unito.it (M.C.D.S.); jeanpiero.margaria@unito.it (J.P.M.); miriam.martini@unito.it (M.M.); fiorella.altruda@unito.it (F.A.); 2Molecular Biotechnology Center, Via Nizza 52, 10126 Torino, Italy; alessandro.giorgio@bioclarma.com (A.G.); giorgia.mandili@bioclarma.com (G.M.); davide.barberio@unito.it (D.B.); 3Department of Clinical and Biological Sciences, University of Torino, 10043 Orbassano, Italy; 4Department of Public Health and Pediatrics, Microbiology and Virology Unit, Azienda Ospedaliero Universitaria “Città della Salute e della Scienza”, University of Torino, 10126 Torino, Italy; paolo.bottino@unito.it (P.B.); elisa.zanotto@unito.it (E.Z.); rossana.cavallo@unito.it (R.C.)

**Keywords:** SARS-CoV-2, COVID-19, qRT-PCR, multiplex qRT-PCR, direct multiplex qRT-PCR

## Abstract

Rapid and sensitive screening of severe acute respiratory syndrome coronavirus 2 (SARS-CoV-2) is essential to limit the spread of the global pandemic we are facing. Quantitative real-time reverse transcription-polymerase chain reaction (qRT-PCR) is currently used for the clinical diagnosis of SARS-CoV-2 infection using nasopharyngeal swabs, tracheal aspirates, or bronchoalveolar lavage (BAL) samples. Despite the high sensitivity of the qRT-PCR method, false negative outcomes might occur, especially in patients with a low viral load. Here, we developed a multiplex qRT-PCR methodology for the simultaneous detection of SARS-CoV-2 genome (*N* gene) and of the human *RNAse P* gene as internal control. We found that multiplex qRT-PCR was effective in detecting SARS-Cov-2 infection in human specimens with 100% sensitivity. Notably, patients with few copies of SARS-CoV-2 RNA (<5 copies/reaction) were successfully detected by the novel multiplex qRT-PCR method. Finally, we assessed the efficacy of multiplex qRT-PCR on human nasopharyngeal swabs without RNA extraction. Collectively, our results provide evidence of a novel and reliable tool for SARS-CoV-2 RNA detection in human specimens, which allows the testing capacity to be expanded and the RNA extraction step to be bypassed.

## 1. Introduction

Severe acute respiratory syndrome coronavirus 2 (SARS-CoV-2), a novel coronavirus, was recently identified in patients with acute respiratory disease and declared by the World Health Organization (WHO) in March 2020 as global pandemic [1] (https://www.who.int/emergencies/diseases/novel-coronavirus-2019). In the absence of specific therapeutic treatments and because of the high infectivity, early-stage detection is essential, especially in asymptomatic or low copy-number patients, to immediately isolate infected people from the healthy population. However, several problems have emerged in clinic diagnostics, particularly in scaling up daily tests and managing the lack of supplies for extraction and evaluation reagents [2]. This situation promoted the search for alternative protocols to ensure continuity in samples screening. Quantitative real-time reverse transcription-polymerase chain reaction (qRT-PCR) is widely used in virology diagnostics to detect causal viruses from respiratory secretions [3]. The Centers for Disease Control and Prevention (CDC) in the United States rapidly developed a qRT-PCR protocol to specifically detect SARS-CoV-2 infection in patients. This protocol is based on separate qRT-PCR reactions targeting the SARS-CoV-2-specific *N* gene [4]. In this study, we reported the development of a multiplex qRT-PCR assay to detect this new virus in human clinical specimens. Two regions of the *N* gene were used to detect SARS-CoV-2 RNA through one-step qRT-PCR. The human *RNase P* gene was amplified in parallel as an internal control to assess the quality of sample collection and to demonstrate successful recovery of the nucleic acid. Notably, the multiplex assay was capable to detect positive samples from nasopharyngeal swabs with 100% sensitivity. We also found that multiplex qRT-PCR assay was effective in detecting SARS-CoV-2 infection, similarly to singleplex qRT-PCR assays performed on the same target sequences. Finally, we investigated the detection efficacy of the multiplex qRT-PCR assay on swab transport medium (termed diluent hereafter) without RNA extraction.

## 2. Materials and Methods

### 2.1. Primers and Probes

The primers were designed to specifically amplify three target regions: the two SARS-CoV-2-specific fragments *N1* and *N2*, and the human *RNAse P* gene. Primers sequences are described in Table 1. N1 and N2 primer/probe sets were designed based on the Atlanta CDC protocol revision dated 24 January 2020 (https://www.cdc.gov/coronavirus/2019-ncov/lab/rt-pcr-panel-primer-probes.html). N1 and N2 primer/FAM-labelend sets belonged to the list of acceptable lots for testing under CDC’s Antiviral Emergency Use Authorization (EUA) (2019-nCoV CDC Probe and Primer Kit for SARS-CoV-2, LGC, Biosearch Technologies, Novato, CA, USA). The human *RNase P* primer/HEX-labeled probe set was purchased from Metabion (Planegg/Steinkirchen, Germany). The 4 × primer/probes mixes were prepared at the following concentrations: SARSCoV-2-specific fragments *N1* and *N2*, 500 nM of primers and 125 nM of probe; human *RNase P* gene, 150 nM of primers, and 100 nM of probe.

### 2.2. Control Samples

Known copies of synthetic DNA plasmids for the two SARS-CoV-2-specific fragments *N1* and *N2* were used to quantify SARS-CoV-2 viral copy number in human samples. A synthetic DNA plasmid for the human *RNase P* gene was used to quantify the *RNase P* gene copy number in human samples. Plasmids were purchased from IDT (Integrated DNA Technologies, Leuven, Belgium). Plasmids were linearized by digesting with ECORI/SWAI. RNA extracted from the HL60 cell line was used as a human specimen control (HSC). The human *RNase P* gene was amplified in parallel to SARS-CoV-2-specific targets as an internal control for nucleic acid extraction procedure as well as to demonstrate the success of nucleic acid recovery in swab diluents that did not undergo RNA extraction. Samples were considered positive to SARS-CoV-2 infection when a signal was detected at Ct < 40 for SARS-CoV-2-specific fragments *N1* and *N2* combined reactions (FAM channel). A sample was considered negative to SARS-CoV-2 infection if the N1/N2 FAM signal was undetermined or detected at Ct > 40, and additionally, the *RNase P* internal control was amplified with a Ct < 35 (VIC channel). A specimen was considered invalid when both N1/N2 signal and RNase P signal were undetermined or detected as negative, according to the aforementioned criteria. A no template control (NTC) consisting of nuclease-free water was used as negative control.

### 2.3. Sample Preparation and Viral RNA Extraction

Clinical samples were obtained from patients with symptoms related to SARS-CoV-2 infection, admitted to the Emergency Department (ED) of University Hospital “Città della Salute e della Scienza di Torino” (Torino, Italy). A nasopharyngeal swab was collected from each patient and analyzed by the Microbiology and Virology Unit of the aforementioned hospital. Samples were collected in 3 mL UTM^®^ swabs (Copan, Brescia, Italy). Viral RNA extraction was performed using an EZ1^®^ Advanced XL instrument (Qiagen, Germantown, MD, USA) with EZ1^®^ DSP Virus Kit (Qiagen, Germantown, MD, USA); starting volume was 200 µL, while elution volume was 90 µL. For the direct RT-PCR, 100 µL of UTM diluents swab from primary samples were heated at 95 °C for 10 min and then cooled at 4 °C. Subsequently, samples were immediately amplified.

### 2.4. Singleplex and Multiplex qRT-PCR

A 7500 Fast Dx Real Time PCR instrument (Applied Biosystems, Waltham, MA, USA) was used for qRT-PCR. Each 20 µL reaction mixture contained 10 µL of 2× qScript XLT One-Step RT-qPCR ToughMix (Quantabio, USA), 5 µL of primers/probes mix, and 5 µL of extracted RNA. Direct multiplex qRT-PCR was performed on a c1000 touch™ CFX 96 Thermal Cycler (Bio-Rad, Irvine, CA, USA) using 10 µL of 2 × qScript XLT One-Step RT-qPCR ToughMix (Quantabio, Beverly, MA, USA), 5 µL of primers/probes mix, and 5 µL of previously heated nasopharyngeal swab diluent. The final concentrations of primers and probes were 500 nM (viral targets primers), 125 nM (viral targets probes), 150 nM (RNase P primers), and 100 nM (RNase P probe). The cycling program was as follows: reverse transcription at 50 °C for 10 min; polymerase activation at 95 °C for 3 min; and 45 cycles of PCR at 95 °C for 3 sec and 55 °C for 30 sec (signal acquisition). The filter combinations were 465–510 (FAM), 540–580 (HEX). Clinical sample manipulations and qRT-PCR assays were performed in a Class IIA Biosafety Cabinet using biosafety level-2 (BSL-2) precautions.

## 3. Results

### 3.1. Comparison between Singleplex and Multiplex One-Step qRT-PCR

To compare the single-target (singleplex) and multitarget (multiplex) assay, RNA samples from 10 SARS-CoV-2 positive patients (Allplex™ 2019-nCoV kit, Seegene Inc., Seoul, Korea) were analyzed simultaneously with the two described methods (Figure 1), evaluating the two viral target regions, *N1* and *N2*. In addition, the human *RNase P* gene was included as internal control to evaluate the quality of clinical specimens (i.e., nasopharyngeal swabs) and nucleic acid extraction.

In the singleplex assay, the three target sequences (i.e., *N1*, *N2, RNase P*) were separately amplified, while in the multiplex assay they were amplified together in a single reaction tube. To this end, the multiplex qRT-PCR was performed using specific primer sets in combination with spectrally distinct fluorescent probes for the human (HEX) and the viral (FAM) targets. Singleplex and multiplex qRT-PCR were performed using the Quantabio qScript XLT One-Step RT-qPCR ToughMix. Thermal cycling conditions were as CDC validated procedures.

The multiplex qRT-PCR displayed 100% sensitivity (Figure 1) and 100% specificity (i.e., true negative rate, data not shown) by correctly identifying all patients with and without the infection, respectively. Interestingly, one positive sample (patient 6) incorrectly resulted as negative with the N1 primer/probe set, thus reducing the epidemiological sensitivity of that reaction. Notably, the same sample was correctly detected as positive to SARS-Cov2 infection by using both the N2 primer/probe set and the triplex reaction. This analysis demonstrates that the multiplex qRT-PCR amplification is effective in detecting positive samples similarly to the N1 and N2 singleplex assays (Figure 1).

### 3.2. Preparation of Standard Curves for Viral RNA Quantification in SARS-CoV-2 Samples

In order to perform absolute quantification of SARS-Cov-2 RNA copy number in COVID-19 samples, purified plasmid DNA was used to prepare the standard curves. In particular, N1/N2 and RNase P reference plasmids were diluted to four concentrations and analyzed by multiplex qRT-PCR. The results showed the correlation between Ct values and plasmid copy number, ranging from 10^2^ to 10^5^ copies/reaction (Figure 2). The regression correlation coefficients of the standard curves were 0.999 for both N1/N2 and RNase P.

### 3.3. Validation of the Multiplex Methods for SARS-CoV-2 Detection and Absolute Quantification

To validate the multiplex qRT-PCR methods, 27 SARS-CoV-2-positive samples and 24 negative samples were analyzed (Table 2). To obtain an absolute quantification of viral copy number in human samples, we included plasmid standards in the same experimental plate. The analysis revealed that multiplex assay displayed 100% true positive and true negative rates on 51 tested human patients (Table 2).

### 3.4. SARS-CoV-2 Detection by Direct Multiplex qRT-PCR without RNA Extraction

The current diagnostic testing methods recommended by CDC in the United States and the World Health Organization (WHO) for testing SARS-CoV-2 require two steps: (a) RNA extraction from patient nasopharyngeal swabs; and (b) qRT-PCR amplification of viral targets in the extracted RNA. The RNA extraction step represents a major bottleneck due to reagents shortage, cost, and time-consuming procedures. To address this issue, we tested the efficacy of the multiplex qRT-PCR method to directly detect viral genes in swab diluent from human patients without extracting RNA. In particular, we compared results from multiplex qRT-PCR performed on RNA extracted from nasopharyngeal swabs and multiplex qRT-PCR performed on not-extracted swab diluents (Figure 3). Overall, 17/20 samples derived from human patients that were positive to SARS-CoV-2 infection were also scored by the direct method (Figure 3), thus, indicating a sensitivity of 85%. Notably, direct multiplex qRT-PCR was always able to detect the internal control (human *RNase P* gene) in nasopharyngeal swab diluents (data not shown).

## 4. Discussion

In the present study, a multiplex qRT-PCR method for the early diagnosis of COVID-19 was developed and optimized. One-step qRT-PCR diagnostic sensitivity is influenced by multiple factors, such as the selection of the target genes, the design of the primer and probe sets and their concentration, the efficiency of the enzymes used, and the method of extraction and storage media [5]. Therefore, validation experiments are necessary to optimize this method for detection of viral RNA in human specimens [6]. The reported multiplex qRT-PCR method is based on the combined multitarget amplification of two fragments (*N1* and *N2*) of the SARS-CoV-2-specific *N* gene. Moreover, human *RNase P* sequence was included as internal control. The results of the study demonstrated that the combined amplification of *N1* and *N2* sequences through multiplex qRT-PCR was sensitive in detecting viral RNA in a similar way to both N1 and N2 detection in separate singleplex qRT-PCR assays. Indeed, *N1* and *N2* SARS-CoV-2-specific fragments were amplified at earlier Ct values using multiplex qRT-PCR compared to singleplex qRT-PCR.

Moreover, the reported method of multiplex qRT-PCR, which is based on the amplification of *N1/N2/RNase P* sequences, showed a similar efficiency to the reference multitarget method (*E* gene of all Sarbecovirus, *N*, and *RNA-dependent RNA polymerase* genes specific for SARS-CoV-2 virus) developed to detect SARS-CoV-2 in human samples. Hence, this method can be considered a valid diagnostic tool, thus providing the advantage of a less complex and unambiguous results interpretation pattern. In addition, the detection of an endogenous human gene as internal control rather than exogenous RNA or DNA standards (provided with other detection methods) enables the correct evaluation not only of the quality of extraction and qRT-PCR reaction, but of the swab sampling as well.

Overall, our multiplex qRT-PCR methodology displayed 100% sensitivity in viral RNA detection in 27 patients who had been previously diagnosed positive to SARS-CoV-2 infection. Moreover, the specificity (i.e., the true negative rate) was 100%, based on the analysis of 24 negative human samples. Interestingly, correlation between viral load and Ct values was done using control plasmids with known copy numbers. Subsequently, using the generated standard curves, the viral copy numbers in the clinical specimens were quantified based on the Ct values. Notably, the use of multiplex assays strongly increases the testing capacity of diagnostic laboratories and can be very important to counteract pandemic emergency.

Additionally, we assessed the feasibility of direct multiplex qRT-PCR on swabs diluents, bypassing the step of RNA extraction. Direct multiplex qRT-PCR showed similar efficiency to that including RNA extraction in detecting positive samples, indicating that RNA extraction could be avoided. In particular, 17/20 positive swab diluents were detected by direct multiplex qRT-PCR. Notably, the three undermined samples displayed high Ct values (Ct > 34) that, according to our quantitative analyses, correspond to less than 5 viral copies.

Notably, on May 27, 2020, the WHO published updated interim guidance on the clinical management of COVID-19, and provided updated recommendations on the criteria for discharging patients from isolation. The updated criteria reflect the recent finding that patients whose symptoms have resolved may still test positive for the COVID-19 virus (SARS-CoV-2) by qRT-PCR for many weeks. Although viral RNA can be detected by PCR with a low limit of detection (LOD), and even after the resolution of symptoms, the amount of detected viral RNA is substantially reduced over time and generally below the threshold at which replication-competent virus can be isolated by culture methods [7,8]. These findings suggest limited clinical relevance in the use of qRT-PCR alone as a decision factor for patients’ isolation, especially for specimens displaying low viral loads, thus suggesting the suitability of our direct extraction-free method. Indeed, our direct multiplex qRT-PCR method showed an acceptable diagnostic sensitivity on specimens with a significant viral load, whereas undetected samples displayed a viral load< 5 copies/reaction. In particular, the lowest Ct among discordant samples (i.e., sample #16) was 34.06 (as detected from purified RNA). According to our standard curve, such a Ct value likely corresponds to approximatively 4 copies/reaction. These findings, taken together, indicate that direct multiplex qRT-PCR represents a valid diagnostic tool for patients’ management and for rapid and low-cost screening.

The direct multiplex qRT-PCR method offers numerous advantages, such as the ability to quickly test symptomatic COVID-19 patients arriving at the hospital emergency room. Moreover, the direct testing would allow saving time and reducing costs. Indeed, the direct protocol is simple and fast, and was demonstrated to be effective in more than 90% of the analyzed samples. Results can be obtained in 70 min without using either extraction or sample pretreatment reagents and plastics. Several aspects can be improved to increase the sensitivity of direct multiplex qRT-PCR. For example, different media for swab collection can be tested to increase the detection efficiency of low copy-number targets and the overall sensitivity.

## 5. Conclusions

In conclusion, the multiplex qRT-PCR methodology displayed high sensitivity in the detection of SARS-CoV-2 RNA. The use of this method to screen human samples during the COVID-19 pandemic will reduce the number of false negative outcomes and the reagents cost. In addition, multiplex qRT-PCR will increase the number of analyzable samples in a single experiment, thus reducing the time required by clinical laboratory to complete the expected number of daily tests. Finally, the multiplex qRT-PCR also proved effective as a direct method, opening new possibilities for the rapid identification of COVID-19 positive patients in hospital emergency and in case of a lack of RNA purification tools.

## Figures and Tables

**Figure 1 microorganisms-08-01064-f001:**
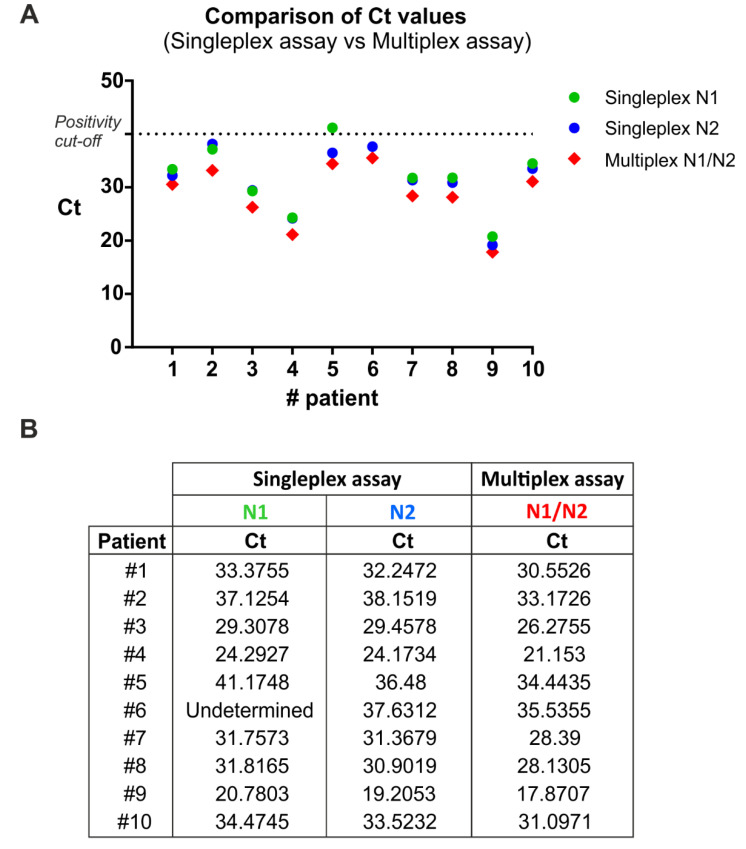
Singleplex and multiplex qRT-PCR comparison. (**A**) Ct values from 10 positive samples were analyzed by qRT-PCR in singleplex and multiplex mode. The exact Ct values are reported in (**B**).

**Figure 2 microorganisms-08-01064-f002:**
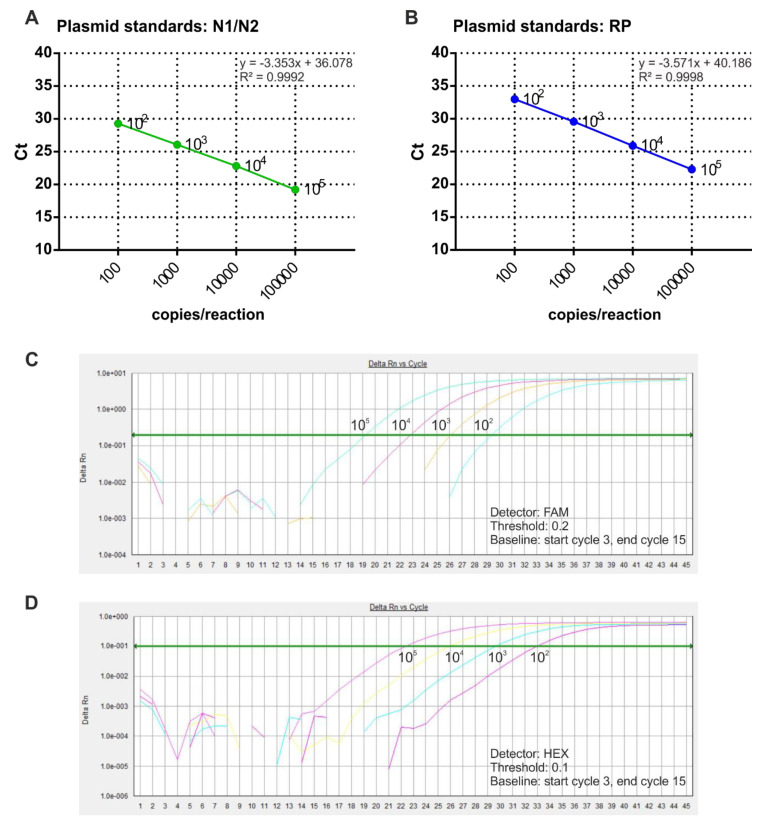
Calibration curves assessment through multiplex RT-qPCR. (**A**,**B**) Sample calibration curves associated withN1/N2 (**A**) and RNase P (**B**) plasmid copy number are shown. The *x*-axis represents the logarithm of the copy number of the plasmid standard, and the *y*-axis represents the Ct value. Amplification curves of different copies of N1/N2 control plasmid (**C**) and RNase P control plasmid (**D**) are shown.

**Figure 3 microorganisms-08-01064-f003:**
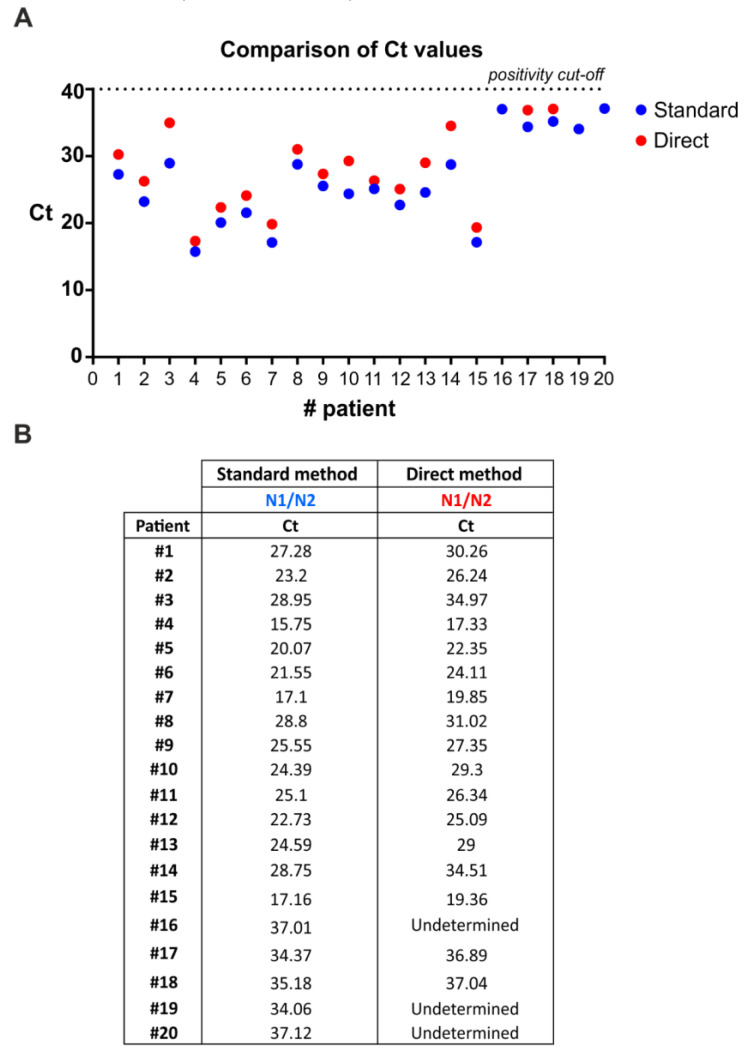
SARS-CoV-2 RNA can be detected from COVID-19 patients’ nasopharyngeal swabs by multiplex qRT-PCR without an RNA extraction step. (**A**) Distribution of Ct values from nasopharyngeal swabs following direct multiplex qRT-PCR versus standard multiplex qRT-PCR that includes RNA extraction. Direct multiplex qRT-PCR performed on 5 µL of nasopharyngeal swab diluent after heating for 10 min at 95 °C. Exact Ct values are reported in (**B**).

**Table 1 microorganisms-08-01064-t001:** Primers and probes used in singleplex and multiplex quantitative real-time reverse transcription-polymerase chain reaction (qRT-PCR).

Name	Oligonucleotide Sequence (5′>3′)	Label
SARS-CoV-2-N1 Forward primer	GAC CCC AAA ATC AGC GAA AT	none
SARS-CoV-2-N1 Reverse primer	TCT GGT TAC TGC CAG TTG AAT CTG	none
SARS-CoV-2-N1 Probe	FAM-ACC CCG CAT TAC GTT TGG TGG ACC-BHQ1	FAM, BHQ-1
SARS-CoV-2-N2 Forward primer	TTA CAA ACA TTG GCC GCA AA	none
SARS-CoV-2-N2 Reverse primer	GCG CGA CAT TCC GAA GAA	none
SARS-CoV-2-N2 Probe	FAM-ACA ATT TGC CCC CAG CGC TTC AG-BHQ1	FAM, BHQ-1
RNAse P Forward primer	AGA TTT GGA CCT GCG AGC G	none
RNAse P Reverse primer	GAG CGG CTG TCT CCA CAA GT	none
RNAse P Probe	FAM-TTC TGA CCT GAA GGC TCT GCG CG-BHQ1	FAM, BHQ-1
RNAse P Forward primer	AGA TTT GGA CCT GCG AGC G	none
RNAse P Reverse primer	GAG CGG CTG TCT CCA CAA GT	none
RNAse P Probe	HEX-TTC TGA CCT GAA GGC TCT GCG CG-BHQ1	Hex, BHQ-1

**Table 2 microorganisms-08-01064-t002:** Severe acute respiratory syndrome coronavirus 2 (SARS-CoV-2) RNA detection through multiplex RT-qPCR performed on RNA samples extracted from human nasopharyngeal swabs. The estimated viral load of the samples not included in the linearity range was reported in *italics*.

	Sample	N1/N2 Ct	Copy Number	RNase P Ct	Copy Number
**Positive Patients**	1	23.86	4416.36	25.16	16,077.85
	2	16.6	*647,480.38 **	23.62	43,402.42
	3	26.87	555.35	24.98	18,105.63
	4	26.24	859.61	27.55	3457.22
	5	21.79	18,328.93	26.44	7084.59
	6	25.65	1290.53	29.47	1003.7
	7	27.33	406.97	25.01	17,789.8
	8	33.33	*6.57 ***	27.12	4569.92
	9	22.87	8669.01	26.59	6413.21
	10	24.5	2833.49	27.77	3002.97
	11	19.25	*104,766.80 **	25.73	11,143.83
	12	18.54	*171,029.16 **	25.57	12,377.84
	13	27.1	475.99	27.45	3680.48
	14	26.36	790.29	28.88	1465.76
	15	33.99	*4.18 ***	30.22	619.45
	16	24.37	3094.90	24.88	19,329.07
	17	27.72	310.41	28.83	1518.03
	18	17.05	*474,185.25 **	26.25	7970.26
	19	26.85	563.10	24.8	20,382.7
	20	25.16	1802.91	26.43	7131.23
	21	26.8	585.33	27.24	4225.18
	22	31.93	*17.15 ***	28.71	1632.51
	23	33.63	*5.36 ***	29.61	915.44
	24	18.89	*134,101.34 **	24.95	18,482.08
	25	29.24	109.51	28.66	1685.03
	26	23.51	5595.00	25.07	17,046.25
	27	33.49	*5.88 ***	28.01	2574.54
**Negative Patients**	1	Undetermined	N/A	27.13	4516.19
	2	Undetermined	N/A	23.49	47,189.11
	3	Undetermined	N/A	27.2	4330.41
	4	Undetermined	N/A	25.63	11,913.58
	5	Undetermined	N/A	26.93	5165.96
	6	Undetermined	N/A	26.8	5589.4
	7	Undetermined	N/A	27.37	3875.15
	8	Undetermined	N/A	25.02	17,656.47
	9	Undetermined	N/A	29.27	1137.76
	10	Undetermined	N/A	26.09	8882.14
	11	Undetermined	N/A	25.01	17,782.67
	12	Undetermined	N/A	26.48	6903.96
	13	Undetermined	N/A	25.87	10,229.2
	14	Undetermined	N/A	28.56	1795.65
	15	Undetermined	N/A	26.76	5741.41
	16	Undetermined	N/A	26.12	8658.5
	17	Undetermined	N/A	23.38	50,847.74
	18	Undetermined	N/A	27.14	4488.83
	19	Undetermined	N/A	25.86	10,249.45
	20	Undetermined	N/A	27.88	2792.39
	21	Undetermined	N/A	22.81	73,221.56
	22	Undetermined	N/A	26.41	7193.5
	23	Undetermined	N/A	26	9409.13
	24	Undetermined	N/A	24.92	18,878.14
**Internal Control**	**HL60 RNA** (5 ng)	Undetermined	N/A	29.53	962.78
**Negative Controls**	**H_2_0**	Undetermined	N/A	Undetermined	N/A
	**H_2_0**	Undetermined	N/A	Undetermined	N/A
**Reporter**		*FAM/FAM*		*HEX*	

* >100,000 copies/reaction; ** <100 copies/reaction.

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
