# Peer review of "A Novel Multiplex qRT-PCR Assay to Detect SARS-CoV-2 Infection: High Sensitivity and Increased Testing Capacity"

_microorganisms, 2020, doi:10.3390/microorganisms8071064_

Round 1

Reviewer 1 Report

Review of ms #860284 submitted by Pterillo et al. to Microorganisms (MDPI) for publication.

In this manuscript, Petrillo et.al describe a novel multiplex RT-PCR assay to detect SARS-CoV-2 infection with high sensitivity and testing capacity. The authors describe a  multiplex RT-PCR assay to detect both SARS-CoV-2 N gene and human RNAse P gene simultaneously in clinical samples with or without RNA isolation.

The following are the critiques to this manuscript.

  1. To compare efficiency of singleplex vs multiplex, the authors should be using equivalent amounts of RNA and not volume.
  2. How do the authors explain the decreased Ct values for many multiplex reactions compared to the single plex reaction? The only rationale for that happening is that the primer probes in the multiplex assays are cross amplfying. This needs to be validated independently.
  3. Traditionally, Ct values >36 have been considered spurious amplifications. Thus, analysis should be restricted to those reactions with Ct <36.
  4. The standard curves used to calculate genome equivalents use dilutions from 10 copies to 105 and are then used to extrapolate copy numbers for many samples. This is inaccurate. The copy numbers of samples must fall within the standard curve of the plasmids used.
  5. The efficiency of the reaction and slope of the standard curve should be shown to ensure copy numbers were calculated accurately.
  6. One major critique is the reproducibility. How many times were these reactions repeated to ensure that the data obtained are not one off but rather reproducible.
  7. The title of the paper should be multiple qRT-PCR and not RT-PCR because the authors use real time / quantitative RT-PCR for detection.
  8. Differences in Ct values are shown however in the absence of replicate data, no statistics can be calculated / discerned. These assays need to be repeated in the samples sets used in the sttudy at least three times by independent researchers before the claim of increased sensitivity can be construed to be true.
  9. What is the swab diluent used? At what time point post swab dilution were samples processed for direct qRT-PCR? These methodological details are likely to significantly alter the outcome of these assays as observations from multiple groups attempting direct qRT-PCR from swabs / other biofluids show rapid degradation of viral RNA in biofluids. While all the direct RT-PCR have Ct values increased by 2 compared to those from direct RNA, these details are important for reproducibility.

Reviewer 2 Report

In their manuscript, Petrillo et al. developed and optimized a multiplex RT-PCR method for the early diagnosis of COVID-19. In general, the manuscript is well-written but contains serious flaws that need to be addressed prior to considering for publication.   Major comments: 1.  This manuscript is limited in scope as previously published or commercially available primers and probes we used in the study. When the authors say “more sensitive” based on the data from 10 clinical specimens without statistical analyses, these conclusions are not serious. It seems obvious and needs no proofs that the more targets are used to detect viral RNA, the higher the sensitivity of the method. The more appropriate design would include testing of a representative panel of specimens from patients with COVID-19-like symptoms with subsequent validation using commercially available test systems. 2.  At least, some statistical analysis should be performed to prove that the multiplex assay is more sensitive than a singleplex assay. 3.  The authors used clinical material only from patients with COVID-19 confirmed by another commercially available test system, which means that the sensitivity of the new assay cannot be higher than the sensitivity of the commercial test system. 4.  While the authors suggest using direct RT-PCR for SARS-CoV-2 RNA detection, the sensitivity 85% seems somewhat low, and this optimization doesn’t seem clinically relevant, especially if the viral load is low and the patient is asymptomatic. Since the chances for folse-negative results increase, more testing will be required to prove that the virus is absent, which eventually will increase the overall costs. Please comment. 5.  The human RNA is used  as an internal control to evaluate the quality of swabbing procedures. But it is not clear how this knowledge can be applied in a clinical setting. Why the Ct threshold for this control is set up as 35, while the Ct cut-off for SARS-CoV-2 RNA is 40? Will the samples with Ct>35 be treated as invalid and the patient will be swabbed again? 6.  Lanes 188-189. The statement “Interestingly, the multiplex RT-qPCR assay allowed quantifying the viral copy number in each of the human samples, thus providing a correlation between viral load and Ct values” is confusing. Correlation between viral load and Ct values was done using control plasmids with known copy numbers. Then, using the generated standard curves the viral copy numbers in the clinical specimens were quantified based on the Ct values.    Minor comments: 1.  No ethical statement was provided. Did the participants sign an informed consent form? Were the specimens from different paragraphs from the same patients? It is not clear how many samples were studied in total. 2.  There is some confusion in FAM/HEX labels in the Table 1. 3.  Figures 1 and 3. The A and B parts basically repeat each other; the part “B” seems redundant. 4.  Figure 3. Several dots are at Ct value = 0. Please correct. 5.  Lanes 92-94. It is not clear how the direct multiplex reaction was set up. The reaction mixture should be given in more details.

Round 2

Reviewer 1 Report

The authors have made improvements to the manuscript as requested and the revisions  are supported by data shown.

One minor change is requested.

Line 118- Please either show the specificity data (data that show that the primer-probes used in this manuscript DO NOT amplify other viral templates, e.g SAR-COV1) or remove the statement on specificity. 

Author Response

We thank the reviewer for his/her comments. We revised our manuscript accordingly. These changes have been highlighted within the revised manuscript.

The authors have made improvements to the manuscript as requested and the revisions are supported by data shown. One minor change is requested.

Line 118- Please either show the specificity data (data that show that the primer-probes used in this manuscript DO NOT amplify other viral templates, e.g SAR-COV1) or remove the statement on specificity.

Thank you for your comment. At that point, we stated that the multiplex qRT-PCR displayed 100% specificity referring to the fact that the true negative rate was 100%. The true negative rate measures the proportion of actual negatives that are correctly identified as such (e.g., the percentage of SARS-Cov-2 negative people who are correctly identified as not having the condition). Nevertheless, we agree with the reviewer that the term “specificity” in this context might give rise to misinterpretation. For this reason, we modified the text to clarify what is meant (line 123).

We used primer/probe sets included in the list of primer and probe materials that are considered acceptable alternatives to the CDC official primer and probe set. CDC has previously assessed the Specificity/Exclusivity of the used primer/probe sets.

In particular, the CDC guidelines report the following information.

2019-nCoV_N1 Assay:

Probe sequence of 2019-nCoV rRT-PCR assay N1 showed high sequence homology with SARS coronavirus and Bat SARS-like coronavirus genome. However, forward and reverse primers showed no sequence homology with SARS coronavirus and Bat SARS-like coronavirus genome. Combining primers and probe, there is no significant homologies with human genome, other coronaviruses or human microflora that would predict potential false positive rRT-PCR results.

2019-nCoV_N2 Assay:

The forward primer sequence of 2019-nCoV rRT-PCR assay N2 showed high sequence homology to Bat SARS-like coronaviruses. The reverse primer and probe sequences showed no significant homology with human genome, other coronaviruses or human microflora. Combining primers and probe, there is no prediction of potential false positive rRT-PCR results.

In summary, the 2019-nCoV rRT-PCR assay N1 and N2, designed for the specific detection of 2019-nCoV, showed no significant combined homologies with human genome, other coronaviruses, or human microflora that would predict potential false positive rRT-PCR results.

The list of acceptable alternative lots of primer and probe materials is available at the following link: https://www.cdc.gov/coronavirus/2019-ncov/downloads/rt-pcr-panel-primer-probes.pdf

Reviewer 2 Report

The authors revised their manuscript according to the reviewers' comments. The responses seem stisfactory, however the discussing point (response to the major comment #4) should be reflected in the manuscript as well (i.e. in the Discussion section). Another suggestion is to emphasize that the “internal control” RNAse P is used to control the quality of sample collection, but not the control of the reaction itself. 

Author Response

We thank the reviewer for his/her comments. We revised our manuscript accordingly. These changes have been highlighted within the revised manuscript.

The authors revised their manuscript according to the reviewers' comments. The responses seem stisfactory, however the discussing point (response to the major comment #4) should be reflected in the manuscript as well (i.e. in the Discussion section).

Another suggestion is to emphasize that the “internal control” RNAse P is used to control the quality of sample collection, but not the control of the reaction itself.

Thank you for your suggestions. We better explained the use of RNAse P in the Introduction (lines 49-50) and in the “Material and Methods” section (lines 75-78). In addition, we added the description of the No Template Control (water) that was missing in the previous version of the manuscript (lines 84-85). Moreover, we added a paragraph within the “Discussion” section to highlight the issue discussed in our previous response to the major comment #4 (lines 211-226).